# FISHER CONTRASTIVE LEARNING: A ROBUST SOLUTION TO THE FEATURE SUPPRESSION EFFECT

## ABSTRACT

Self-supervised contrastive learning (SSCL) is a rapidly advancing approach for learning data representations. However, a significant challenge in this paradigm is the feature suppression effect, where useful features for downstream tasks are suppressed due to dominant or easy-to-learn features overshadowing others crucial for downstream performance, ultimately degrading the performance of SSCL models. While prior research has acknowledged the feature suppression effect, solutions with theoretical guarantees to mitigate this issue are still lacking. In this work, we address the feature suppression problem by proposing a novel method, Fisher Contrastive Learning, which unbiasedly and exhaustively estimates the central sufficient dimension reduction function class in SSCL settings. In addition, the embedding dimensionality is not preserved in practice. FCL empirically maintains the embedding dimensionality by maximizing the discriminative power of each linear classifier learned through Fisher Contrastive Learning. We demonstrate that using our proposed method, the class-relevant features are not suppressed by strong or easy-to-learn features on datasets known for strong feature suppression effects. Furthermore, we show that Fisher Contrastive Learning consistently outperforms existing benchmark methods on standard image benchmarks, illustrating its practical advantages.

## 1 INTRODUCTION

Among various approaches to self-supervised learning, self-supervised contrastive learning (SSCL) Chen et al. (2020a); Robinson et al. (2020); Kalantidis et al. (2020); Grill et al. (2020); Radford et al. (2021); Chen et al. (2020b;c); Caron et al. (2020); He et al. (2020); Chen & He (2021); Grill et al. (2020) has emerged as a particularly promising technique. SSCL offers a new paradigm that exploits data augmentations to create positive and negative pairs for learning data point representations. For example, in the context of computer vision, positive and negative pairs are formed based on image augmentation techniques such as cropping, color jittering, and adding noise. The augmented image and the original image form a positive pair. Conversely, those images augmented from different images from distinct sources form the negative pairs. The objective of SSCL is to ensure that representations of positive pairs are closer in the embedding space than those of negative pairs. SSCL has achieved remarkable success in various machine learning tasks Chen et al. (2020a); He et al. (2020); Chuang et al. (2020); Radford et al. (2021).

**Feature Suppression Effect** Feature suppression effect is a phenomenon in which dominant features (e.g., content) can overshadow and suppress other important features (e.g., style), causing SSCL to fail to learn the necessary features for downstream tasks Rusak et al. (2022), limiting the potential of SSCL. Feature suppression effects happen when the representations lose diversity and become less informative. For example, easy-to-learn shared features of augmented pairs could suppress the learning of other features Chen et al. (2021), and color features can suppress other features like texture and shape Chen et al. (2020a); Robinson et al. (2021), despite the object class often being determined by features other than just color. Consequently, the presence of "color distribution" suppresses the competing features of "object class" Chen et al. (2021), leading to insufficient dimension reduction space for downstream classification tasks. The resulting lack of discrimination power fails to capture the full complexity and richness of the data, degrading the

performance of downstream tasks. Severe feature suppression effects in SSCL require effective strategies with theoretical guarantees to mitigate this critical problem.

Previous theoretical investigations have shown that under the following scenarios, there would be feature suppression effects: (1) The distribution of the feature is uniformly distributed on the underlying space Robinson et al. (2021); (2) the augmentation tends to preserve class-irrelevant features Xue et al. (2023), and (3) the low embedding dimensionality, meaning the embedding space has a lower rank than its dimension Xue et al. (2023); Li et al. (2023). Based on their understanding of how feature suppression effects happen, they propose several remedies to overcome the feature suppression effect, Robinson et al. (2021) propose implicit feature modification to remove well-represented features in the input samples to encourage the encoder to learn more semantic features. However, the solution works only given the condition that the encoder before feature modification is a shortcut solution. There is no guarantee that the learned features are effective for discrimination. Built on the theoretical understanding, Xue et al. (2023) suggest two pathways to remedy the feature suppression effect, one is to increase the embedding dimensionality, and the other is to prevent imperfect augmentations, i.e. adding noise to class-relevant features. However, they do not offer a robust solution to imperfect augmentations and increase the embedding dimensionality. Besides increasing the embedding dimensionality, predictive contrastive learning Li et al. (2023) prevents the features from being suppressed by training a decoder to restore the input. However, the decoder module also preserves the class-irrelevant features like background information. Although the aforementioned research has alleviated the feature suppression effects by minimizing the information loss in embeddings, methods guaranteeing that the learned features are both effective and sufficient in self-supervised learning are still lacking.

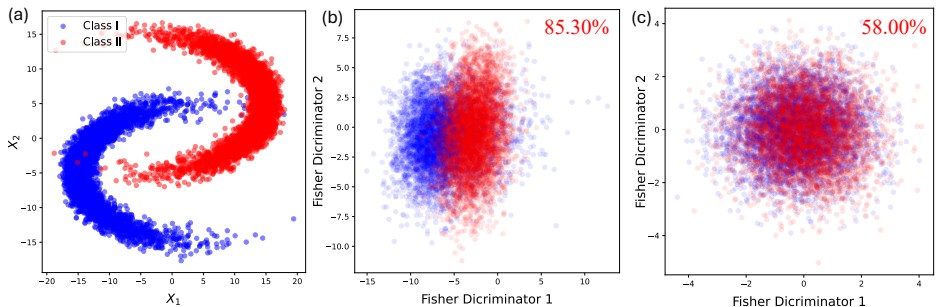

Figure 1: Results for SimCLR and the proposed method on an 18-dimensional self-supervised learning task. (a) The true predictors of data for downstream classification tasks. (b) FCL: The learned embedding space is linear-transformed by discriminant functions of the embedding space. We use the representation learned by FCL to train a K-Nearest Neighbor (KNN) classifier and the accuracy is 85.30%. (c) SimCLR: The learned embedding space is linear-transformed by discriminant functions of the embedding space. We use the representation learned by SimCLR to train a KNN classifier and the accuracy is 58.00%.

**Fisher Discriminant Analysis and Sufficient Dimension Reduction** Fisher discriminant analysis (FDA) is a supervised dimension reduction method that projects data onto a lower-dimensional space while maximizing the separation between different classes Hastie et al. (1994; 1995). FDA as a classification method can be generalized to regression settings, e.g., sliced inverse regression Li (1991); Chen & Li (2001), a method of estimating the sufficient dimension reduction (SDR) subspace Chen & Li (1998); Cook (2007). Based on the theoretical framework of SDR, the discriminant functions in FDA recover the effective dimension reduction subspace for binary responses in the classification tasks Chen & Li (2001). In a general setting, sufficient dimension reduction subspace can be defined as a sub $\sigma$-filed $\mathcal{G}$ of $\sigma(X)$ such that $Y \perp\!\!\!\perp X \mid \mathcal{G}$, where $\perp\!\!\!\perp$ denotes statistical independence. This indicates that $\mathcal{G}$ preserves all the information about $Y$ contained in $X$. SDR has played crucial roles in dimension reduction problems of regression Li (1991); Cook & Li (2002) and classification tasks Chen & Li (2001); Wu (2008). By embedding the data into the sufficient dimension reduction subspace, we can obtain a dimension reduction space that is both unbiased and exhaustive Lee et al. (2013).

**Motivating Example** We illustrate such a phenomenon through an example of classification, introduced by Meng et al. (2020). We adapt it to a self-supervised learning task. The example includes two C-shaped curves with random Gaussian noise in a two-dimensional subspace embedded in $\mathbb{R}^{18}$. There are two classes:

- $X_1 = 20\cos(\theta) + Z_1 + 5$, $X_2 = 10\sin(\theta) + Z_2 - 5$, where $Z_1, Z_2$ and $\theta$ are independently generated from $\mathcal{N}(0,1), \mathcal{N}(0,1)$, and $\mathcal{N}(\pi, (0.25\pi)^2)$, respectively; $X_3, ..., X_{18}$ are independently generated from $\mathcal{N}(0,5)$.
- $X_1 = -20\cos(\theta) - Z_1 - 5$, $X_2 = 10\sin(\theta) + Z_2 + 5$, where $Z_1, Z_2$ and $\theta$ are independently generated from $\mathcal{N}(0,1), \mathcal{N}(0,1)$, and $\mathcal{N}(\pi, (0.25\pi)^2)$, respectively; $X_3, ..., X_{18}$ are independently generated from $\mathcal{N}(0,5)$.

For each class, we first generate a sample of 5000 in size. The 10,000 data points in the first two dimensions are shown in panel (a) of Figure 1. We train the data using a single-layer neural network using the SSCL framework. The dimension of the embedding space is 10. The augmentation for the SSCL is adding Gaussian random noise $\mathcal{N}(0,1)$. The class-relevant features are $X_1$ and $X_2$. However, the easy-to-learn and strong features, $X_3, ..., X_{18}$, suppress the class-relevant features as shown in the panel (c) of Figure 1. We visualize the 2D embedding space of SimCLR by projecting it onto Fisher discriminant directions.

**Fisher Contrastive Learning**    To be more robust to the feature suppression effect, we propose a novel contrastive learning method, Fisher Contrastive Learning (FCL), which estimates the sufficient dimension reduction subspace through nonlinear transformation. First, we reformulate the SSCL problem as a dimension reduction task for classification, where the goal is to project data into a subspace that enhances class separability induced by augmented image views. Second, we maximize the discrimination power of the Fisher discriminant functions in the embedding space. By doing so, we preserve the central sufficient dimension reduction functional class of the self-supervised learning task as shown in panel (b) of Figure 1. Consequently, the embeddings learned by FCL can mitigate the feature suppression effects and retain informative components with theoretical guarantees. Our contributions are threefold: (1) We introduce FCL, a nonlinear sufficient dimension reduction method for SSCL, that offers a robust solution to the feature suppression effects by learning an exhaustive and unbiased function class; (2) FCL prevents the low embedding dimensionality by maximizing the discrimination power of the Fisher discriminant functions; (3) we demonstrate the effectiveness of the proposed method on various datasets that exhibit feature suppression effects and benchmark image datasets compared with other self-supervised learning methods.

## 2 PRELIMINARIES

Simple contrastive learning (SimCLR) Chen et al. (2020a) is one of the most powerful methods in SSCL. Our presentation of SSCL will focus on SimCLR. Suppose that the dataset contains $N$ data points. In SimCLR, we apply augmentations to each input vector $\boldsymbol{x}_k$, where each input vector generates augmented pairs, and we finally get $2N$ data points. The augmented positive pairs of the original input $\boldsymbol{x}_k$ are denoted as $\tilde{\boldsymbol{x}}_{2k-1}$ and $\tilde{\boldsymbol{x}}_{2k}$, randomly sampled from the space $\Omega_X$ and $X$ is a random vector of dimension $p$. We define the non-linear mapping for each data point by: $f : \Omega_X \to \mathbb{R}^d$, transforming the data point from the input space to the embedding space, $\boldsymbol{z}_k = f(\boldsymbol{x}_k)$, where $\boldsymbol{x}_k \in \mathbb{R}^p$, $f(\boldsymbol{x}_i) \in \mathbb{R}^d$, and $d < p$. In other words, the mapping pulls data to the embedding space of dimension $d$. The similarity score $s_{i,j}$ of two images is defined based on the embeddings, $\boldsymbol{z}_i$ and $\boldsymbol{z}_j$. In SimCLR, the pairwise similarity score is cosine similarity, $s_{i,j} = \boldsymbol{z}_i^\top \boldsymbol{z}_j / (\|\boldsymbol{z}_i\| \|\boldsymbol{z}_j\|)$. The goal of SimCLR is to discriminate augmented samples of one image from the augmentations of other images. The contrastive loss function is defined by:

$$\ell(i,j) = -\log \frac{\exp(s_{i,j}/\tau)}{\sum_{k=1}^{2N} \mathbf{1}_{[k \neq i]} \exp(s_{i,k}/\tau)}, \quad \mathcal{L} = -\frac{1}{2N} \sum_{k=1}^{N} [\ell(2k-1, 2k) + \ell(2k, 2k-1)], \quad (1)$$

where $\tau$ is the temperature controlling the hardness of negative samples. The standard contrastive loss can be generalized to the composition of two parts, one is uniformity loss, the other is alignment loss Wang & Isola (2020); Chen et al. (2021).

$$\mathcal{L} = \underbrace{-\frac{1}{N} \sum_{i,j} \text{sim}(\boldsymbol{z}_i, \boldsymbol{z}_j)}_{\mathcal{L}_{\text{alignment}}} + \underbrace{\frac{\lambda}{N} \sum_i \log \sum_{k=1}^{2N} \mathbf{1}_{[k \neq i]} \exp(\text{sim}(\boldsymbol{z}_i, \boldsymbol{z}_k)/\tau)}_{\mathcal{L}_{\text{distribution}}}, \quad (2)$$

where $\mathcal{L}_{\text{alignment}}$ encourages embeddings of augmented pairs to be mapped together, while $\mathcal{L}_{\text{distribution}}$ encourages the augmentations from different samples to spread as much as possible.

## 3 FISHER CONTRASTIVE LEARNING

The occurrence of feature suppression effects in SimCLR deteriorates the performance of downstream tasks. Such issues occur when some features are too easy or strong to learn other discriminant features for downstream tasks. To address the problem, we propose FCL, a novel method that projects the data into the central sufficient dimension reduction function class. In addition, our proposed algorithm can preserve the dimensionality of embeddings, further guaranteeing the embeddings are robust to feature suppression effects.

Since the generalized contrastive loss in Eq. 2 aligns with the goal of the FDA, we first convert the SSCL to a classification task. The classification labels are defined such that the augmentations from the $j$-th data point belong to the $j$-th class. There are $n$ classes for each batch of size $n$. The primary objective of the FDA is to find a hyperplane that optimally separates the classes by maximizing the between-class variance relative to the within-class variance. For high dimensional data, the FDA may not be effective enough for classification tasks Dorfer et al. (2015); Hastie et al. (1995). This motivates us to introduce a nonlinear method, FCL. After a nonlinear mapping, we optimize the FDA objective in the embedding space trained by the neural network. In FCL, the alignment loss corresponds to the within-class variance, and the uniformity loss corresponds to the negative of between-class variance. For each batch of size $n$, we define the within-class variance of the embeddings by:

$$S_W := \frac{1}{2n} \sum_{i=1}^{n} \left(f(\tilde{x}_{2i-1}) - f(\tilde{x}_{2i})\right) \left(f(\tilde{x}_{2i-1}) - f(\tilde{x}_{2i})\right)^{\top}, \tag{3}$$

where $S_W$ is a matrix of dimension $(d, d)$. Within-class variance measures the variance within each class. The goal is to minimize within-class variance over between-class variance, ensuring that the members of each class are as close as possible to their respective class mean. The between-class variance of the embeddings measures how much each class mean differs from the overall mean,

$$S_B := \sum_{i=1}^{n} 2 \left(\mu_i - \mu\right) \left(\mu_i - \tilde{\mu}\right)^{\top}, \tag{4}$$

where $\mu_i = \frac{1}{2} \left(f(\tilde{x}_{2i-1}) + f(\tilde{x}_{2i})\right)$, and $\mu = \frac{1}{2n} \sum_{i=1}^{n} \left(f(\tilde{x}_{2i-1}) + f(\tilde{x}_{2i})\right)$. Same as FDA, the total variance $S_T$ is the sum of the within-class variance and between-class variance $S_T = S_W + S_B$. The objective function of FCL is:

$$\max_{U} \ \text{tr} \left(\frac{U^{\top} S_B U}{U^{\top} S_W U}\right), \tag{5}$$

with respect to the matrix $U$, and the column space of the matrix $U$ consists of the discriminant functions for FCL. Maximizing the above function equals to the problem:

$$\max_{U} \text{tr} \left(U^{\top} S_B U\right), \text{s.t.} \ U^{\top} S_W U = I. \tag{6}$$

To optimize the problem, we can use the Lagrange algorithm, that is, to maximize:

$$\mathcal{L} = \text{tr} \left(U^{\top} S_B U\right) - \text{tr} \left(\Lambda^{\top} \left(U^{\top} S_W U - I\right)\right). \tag{7}$$

The solution to the optimization function is

$$S_B U = S_W U \Lambda, \tag{8}$$

where $U$ is the eigen-matrix of $S_W^{-1} S_B$. The eigenvectors are nonlinear Fisher discriminant functions, also referred to as the canonical variates Chen & Li (2001). The optimization goal of FCL is to maximize the discrimination power defined in Fukunaga (1990), the sum of variances of embeddings projected to the canonical variates:

$$\max_{f} \ \Delta = \text{tr}(S_W^{-1} S_B). \tag{9}$$

Since $\Delta$ is the upper bound of the misclassification error, maximizing the discrimination power inherently minimizes the alignment loss and uniformity loss in contrastive learning Bian & Tao (2014). When implemented, $S_W$ is singular when the data resides in a lower-dimensional space than the dimension of embeddings, $d$. To ensure the numeric stability, we add a regularization term to the within-class variance Zhong et al. (2005), $S_B U = (S_W + \lambda I) U \Lambda$, where $\lambda$ is a penalty hyperparameter selected by grid search.

## 4 CENTRAL $\sigma$-FIELDS FOR NONLINEAR SUFFICIENT DIMENSION REDUCTION

Sufficient dimension reduction (SDR) has been one of the most popular dimension reduction frameworks in statistics (Li, 1991; Cook & Li, 2002; Li, 2018). In the classical setting, linear SDR equals FDA, which seeks a low-dimensional linear classifier that captures all the information needed in a classification task. In the self-supervised contrastive learning framework, we have a random vector $X$ of dimension $p$ comprising augmented data and random variable, a random variable $Y$ augmented classification labels. If there is a matrix $\mathbf{B} \in \mathbb{R}^{p \times d} (p \geq d)$ such that $Y \perp\!\!\!\perp X \mid \mathbf{B}^\top X$, then the subspace spanned by the column space of $\mathbf{B}$ is referred to as a linear SDR subspace. The intersection of all the SDR subspaces is called the central SDR subspace, which can be estimated by FDA.

Our proposed method FCL generalizes FDA to a nonlinear setting, which generalized SDR subspace to SDR $\sigma$-field, preserving the essential features and variations in the augmented data through nonlinear SDR. Functions are typically defined on a $\sigma$-field. A $\sigma$-field ensures that we can perform key operations without leaving the domain of our measure. Similarly, the central SDR $\sigma$-field is the intersection of SDR $\sigma$-field and can be estimated unbiasedly and exhaustively by our proposed FCL. The unbiasedness ensures that the learned embeddings do not contain redundant information to distinguish those data augmented from different data points, while exhaustiveness guarantees that the embeddings capture all the necessary information to distinguish between different data points, thereby preventing feature suppression.

Other contrastive learning methods would suppress class-relevant features when the augmentation is imperfect Xue et al. (2023). However, our proposed FCL can preserve the class-relevant features illustrated by Figure 1. We prove it by showing the unbiasedness and exhaustiveness of the estimated function class by FCL, which indicates the learned embeddings contain sufficient information from the input images, including the class-relevant features. This suggests that our method is more robust against feature suppression effects compared with other contrastive learning methods.

In this section, we first define the SDR $\sigma$-field and central SDR $\sigma$-field and then prove the proposed FCL can learn the central sufficient dimension reduction $\sigma$-field for the self-supervised learning task. The set of all central $\sigma$-field-measurable, square-integrable functions is named the central SDR function class, which is spanned by a vector of functions $\{g_1, \ldots, g_d\}$. The estimated embedding function $f_{sol}$ is unbiased and exhaustive of the central sufficient dimension reduction function class.

**Definition 4.1.** *A sub $\sigma$-field $\mathcal{G}$ of $\sigma(X)$ is a sufficient dimension reduction(SDR) $\sigma$-field if $Y$ is independent of $X$ conditioned on $\mathcal{G}$, i.e.,*

$$Y \perp\!\!\!\perp X \mid \mathcal{G}, \tag{10}$$

*where $\perp\!\!\!\perp$ denotes statistical independence.*

The special property of the nonlinear sufficient dimension reduction framework is its flexibility, as it does not impose any specific assumptions about the relationship between $Y$ and $X$. By leveraging the general concepts of $\sigma$-field, it also does not require a predefined form for the dimension reduction. To aid those unfamiliar with this concept, we provide the following interpretation in the context of linear dimension reduction.

In terms of nonlinear SDR, our goal is to recover the smallest $\sigma$-field satisfying Eq.(10). To ensure the existence and uniqueness of the smallest $\sigma$-field, we need assumptions on the probability measure for generating the augmented data. Suppose the augmentation on the data point $x_i$ is conducted by random sampling from the probability measure $\mathbb{P}_i$.

**Lemma 4.2** (Existence and uniqueness of the central $\sigma$-field). *If $\mathbb{P}_i$s are dominated by a $\sigma$-finite measure, there exists a unique $\sigma$-field denoted by $\mathcal{Y}_{Y|X} \subset \sigma(X)$ such that*

    *1. $Y \perp\!\!\!\perp X \mid \mathcal{Y}_{Y|X}$,*

    *2. if $\mathcal{G}$ is a SDR $\sigma$-field of $\sigma(X)$, $\mathcal{Y}_{Y|X} \subseteq \mathcal{G}$.*

*The $\sigma$-field $\mathcal{Y}_{Y|X}$ is referred to the central $\sigma$-field.*

Based on the above lemma, we may define the central SDR function class that corresponds to the central $\sigma$-field.

**Definition 4.3.** *The central sufficient dimension reduction function class, denoted by $\mathfrak{S}_{Y|X}$, is defined by*

$$\overline{\text{span}} \left\{ g \in \mathcal{H}_X : g \text{ is measurable } \mathcal{Y}_{Y|X} \right\}$$

Notice that there are infinities functions in the central SDR function class $\mathfrak{S}_{Y|X}$. Hence, the estimability of $\mathfrak{S}_{Y|X}$ relies on the following assumption,

**Assumption 4.4.** *There exists functions $g_1 \ldots, g_d \in \mathfrak{S}_{Y|X}$ such that $\mathfrak{S}_{Y|X} = \overline{\text{span}} \left\{ g_1, \ldots, g_d \right\}$.*

Under this assumption, we can see that the function in $\mathfrak{S}_{Y|X}$ is the linear combination of the functions $\{g_1, \ldots, g_d\}$. Therefore, with Assumption 4.4 and Lemma 4.2, we can see that

$$Y \perp\!\!\!\perp X \mid g_1(X), \ldots, g_d(X). \tag{11}$$

Therefore, the objective is to identify a function $f = (g_1, \ldots, g_d) : \Omega_X \to \mathbb{R}^d$ satisfying the above conditions.

**Assumption 4.5.** *$\mathfrak{S}_{Y|X}$ is complete: For each $\mathcal{Y}_{Y|X}$-measurable function $g \in \mathcal{H}_X$ we have:*

$$E(g(X) \mid Y) = 0 \quad \text{almost surely} \quad \Rightarrow \quad g(X) = 0 \quad \text{almost surely.} \tag{12}$$

When a complete SDR class exists, it is unique and coincides with the central SDR function class. We now provide the main theorem of this section.

**Theorem 4.6.** *Under the Assumption 4.4, Assumption 4.5 and several additional assumptions which are listed in the Appendix, the central SDR function class $\mathfrak{S}_{Y|X}$ can be recovered by solving*

$$\max_{f=(g_1,\ldots,g_d)} \text{tr} \left\{ \text{Var}[f(X)]^{-1} \text{Var}[\text{E}(f(X)|Y)] \right\}, \tag{13}$$

*such that the functions $g_1, \ldots, g_d$ are linear independent. Then, the estimate $f_{sol} = (g_{1,sol}, \ldots, g_{d,sol})$ is (1) unbiased, i.e., $g_{1,sol}, \ldots, g_{d,sol} \in \mathfrak{S}_{Y|X}$; (2) exhaustive, i.e., $\mathfrak{S}_{Y|X} \subset \overline{\text{span}} \left\{ g_{1,sol}, \ldots, g_{d,sol} \right\}$. Therefore, the estimate $f_{sol}$ is Fisher consistent.*

The theoretical result suggests the approaches to estimating the central SDR function class. In fact, Eq.(9) is the sample level of Eq.(13).

**Remark.** The unbiasedness of the estimation of the central SDR function class indicates the regression function class, $f_{sol} = (g_{1,sol}, \ldots, g_{d,sol})$, is contained in the central SDR function class. The exhaustiveness of estimation means the regression function class can generate the central SDR function class. In other words, the proposed FCL can preserve the class-relevant features even when the class-irrelevant features are strong and easy to learn through estimating the central SDR class unbiasedly and exhaustively.

## 5 EXPERIMENT RESULTS

To illustrate the advantage of our proposed FCL method, we conduct experiments on several datasets, including datasets with various levels of feature suppression effects and benchmark image datasets. We also compare our method with other self-supervised learning approaches.

### 5.1 EXPERIMENT DATA

We start by examining two datasets that are designed to show feature suppression effects following Chen et al. (2021). This allows us to demonstrate how our proposed FCL method effectively handles such challenging scenarios. Following this, we evaluate FCL's performance on several benchmark datasets. These benchmarks highlight FCL's capability to learn discriminative features across diverse data distributions and downstream tasks.

### 5.1.1 RANDOM BITS

Random Bits dataset Chen et al. (2021) dataset concatenates real images with images of random integers along the channel dimension. The random integers are sampled from the range $[1, \log_2(l)]$, where $l$ is a controllable parameter, and replicated across all pixels using $l$ binary channels. Notably,

unlike RGB channels, these random bit channels remain identical between augmented views of the same image. The added random bit channels act as easy-to-learn features that can suppress other features. We vary the number of random bits $\{0, 2, 4, 6, 8, 10\}$ to control the mutual information between augmented views. More random bits indicate stronger feature suppression effects.

### 5.1.2 Digits on Flower Dataset

Digits on Flower dataset is adapted from Chen et al. (2021). This dataset involves randomly mapping different numbers of MNIST digits LeCun (1998) onto flower images from five classes in ImageNet Deng et al. (2009): dandelions, daisies, tulips, sunflowers, and roses. Figure 2 illustrates examples with $\{0, 1, 4, 9, 16\}$ digits mapped onto the flower images. MNIST digits in the dataset are easy-to-learn features. As more digits overlap the flower images, the feature suppression effects are stronger.

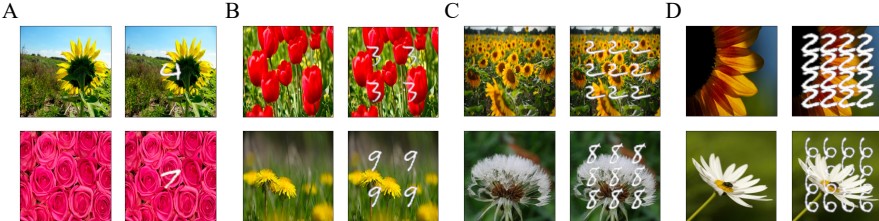

Figure 2: Illustrations of the Digits on Flower dataset. Each group displays different numbers of digits mapped onto flower images. Group A: 1 digit; Group B: 4 digits; Group C: 9 digits; Group D: 16 digits. In each group, the left panel shows the original flower images, while the right panel shows the images after mapping.

### 5.1.3 Benchmark Dataset

We use several image classification benchmark datasets to evaluate proposed algorithm, STL-10 Coates et al. (2011), CIFAR-10, CIFAR-100 Krizhevsky et al. (2009), and Tiny ImageNet mnmoustafa (2017). More details of the description of the datasets can be referred to Section C in Appendix.

### 5.2 Feature Suppression Effects Experiment Results

The experiment results first demonstrate that datasets with strong feature suppression effects degrade the performance of SimCLR and SimSiam for downstream tasks, due to low embedding dimensionality and less discriminative power. In contrast, our proposed FCL method outperforms SimCLR Chen et al. (2020a) and SimSiam Chen & He (2021) on these datasets exhibiting feature suppression effects, showcasing its robustness to such challenging scenarios. For the Random Bits and Digits on Flower dataset experiments, we employed a consistent model architecture: an encoder with three 2D convolutional layers, a flatten layer, and a dense layer, using batch normalization and ReLU activations throughout. The projection network has two dense layers, with a 128-dimensional output.

### 5.2.1 Results of Random Bits Datasets

We compare the proposed method with the benchmark algorithm, SimCLR Chen et al. (2020a) and SimSiam Chen & He (2021), on various levels of feature suppression effects. The accuracy of these three methods is displayed on the left panel of Figure 3, where varying numbers of bits have been added as additional channels invariant in two augmented views. Initially, with a small number of random bits, both methods achieve high classification accuracy for the dataset. However, as the number of random bits in-

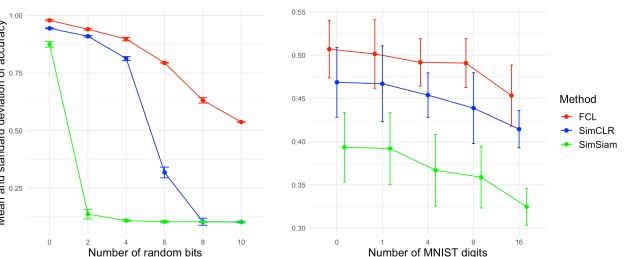

Figure 3: Accuracy for Digits on Flower and Random Bits dataset. Left: Top-1 accuracy for Random Bits dataset. Right: Top-1 accuracy for Digits on Flower dataset. Red lines: FCL method; Blue lines: SimCLR method; Green lines: SimSiam method.

creases, the accuracy of the SimCLR and SimSiam drops sharply. When the number of random bits

reaches 8 and 10, their performance becomes equivalent to random guessing, with an accuracy of about 0.1 for the 10-class dataset. In contrast, our FCL method maintains a higher level of accuracy as the number of random bits increases, staying above 0.5 even when the number of bits reaches 10. The consistent outperformance of FCL underscores its robustness and to feature suppression effects.

We perform the sensitivity analysis of two sets of channels, RGB channels and random bits channels, to further investigate if our proposed method learns more information in random bits relative to RGB channels. First, we perturb the random bits channels by adding different levels of Gaussian random noise to each channel, then measure the difference between the perturbed and original embeddings. Second, we do the same perturbation to the RGB channels and measure the change of embeddings. Third, we use the ratio of change in random bits channels to the change in RGB channels. The results showing the ratio of changes are presented in Table 1. The ratio is larger, and the learned embeddings are more sensitive to adding noise to random bits channels compared with adding noise to RGB channels. More sensitive embeddings indicate more features in RGB channels are suppressed by random bits. Compared with benchmark SSCL methods, SimCLR and SimSiam, our proposed FCL undergoes less ratio of change. The differences of embeddings when adding noise to random bits channels and RGB channels are presented in Table A.3 and A.4, respectively in the Appendix.

In practice, the proposed FCL maximizes the discrimination power and prevents low embedding dimensionality. The left panel of Figure A.2 also shows the rank of the embedding space learned by three SSCL methods. The rank of the embedding space for the SimCLR method varies significantly with different numbers of random bits. In contrast, the rank of the representations learned by the FCL remains stable regardless of the number of random bits.

Table 1: The table shows the ratio of change in extra bits to the change in RGB channels. Different columns correspond to different levels of Gaussian random noise with variances of 0.1, 0.2, 0.3, 0.4, 0.5.

|         | 0.1 | 0.2  | 0.3  | 0.4  | 0.5  |
|---------|-----|------|------|------|------|
| SimCLR  | 0   | 0.41 | 0.82 | 0.89 | 0.92 |
| SimSiam | 0   | 1.14 | 5.16 | 7.95 | 9.90 |
| FCL     | 0   | **0.09** | **0.22** | **0.26** | **0.27** |

### 5.2.2 RESULTS OF DIGITS ON FLOWER DATASET

The accuracy of predicting the label of flowers by SimCLR and SimSiam decreases when the number of digits increases as shown by the blue line and green line in the right panel of Figure 3, respectively. However, the proposed FCL has robust performance even when the feature suppression effects are stronger. To further validate whether our proposed method can effectively learn features for downstream tasks without being overshadowed by digit-related features, we utilize a saliency map for each method to visualize the contribution of each pixel to the prediction, as shown in Figure 4. Compared to SimCLR and SimSiam, the proposed FCL focuses more on the relevant features of the flower rather than the digits. As a result, SimCLR misclassifies the sunflower image (panel (a) in Figure 4) as a dandelion, and SimSiam incorrectly predicts it as a tulip. In another case (from panel (e)-(h) in Figure 4), SimCLR gives more weight to the overlapping digit "two," overlooking important flower features, and consequently predicts the image as the rose. SimSiam also emphasizes the digits and background over the flower's features, leading to a misclassification as a dandelion. Comparing panels (c) and (g), the difference of the SimCLR's salient maps is minor for two different flower images since the digits overshadow other features. However, the dominant features in panels (b) and (f) are different, indicating the proposed FCL learns class-relevant (sunflower) features and is robust to feature suppression effects. More examples are presented in Figure A.1, respectively in the Appendix.

Additionally, the right panel in Figure A.2 shows the rank of the embedding space for each method. This confirms that FCL's embedding dimensionality is more robust to strong feature suppression effects. As the number of digits increases, all methods experience a slight decrease in embedding dimensionality, but FCL maintains a more stable performance.

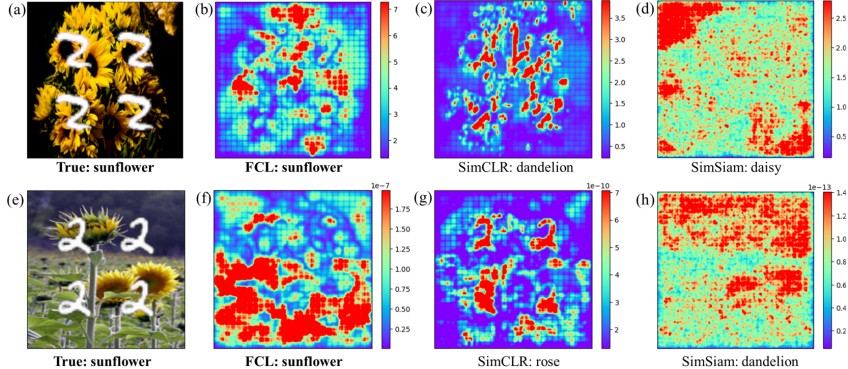

Figure 4: Illustrations of the salient map of each method. The first column is the four-digits-on-flower image. From the second to fourth columns, salient maps of the images for each method, FCL, SimCLR, and SimSiam. The first row shows true and salient maps of one sunflower image, and the second row is another example of sunflower.

## 5.3 BENCHMARK DATASETS EXPERIMENT RESULTS

We compared the proposed FCL with five popular self-supervised learning approaches, SimCLR Chen et al. (2020a), SimSiam Chen & He (2021), MOCO He et al. (2020), BYOL Grill et al. (2020), and DirectCLR Jing et al. (2022) on the benchmark image datasets. The experimental results presented in Table 2 provide a comparative analysis of the linear evaluation accuracy across four benchmark datasets (STL-10, CIFAR-10, CIFAR-100, and Tiny-Imagenet). Notably, FCL demonstrates competitive results with leading accuracies. The proposed FCL is more advantageous than other SSCL methods especially when the task of image classification has more classes, i.e. there are 200 classes in the Tiny ImageNet dataset. The more the number of classes, the more class-relevant features need to be preserved. However, the low embedding dimensionality issue is more severe, which means the class-relevant features could be suppressed for the Tiny ImageNet dataset. We further provide the rank of the covariance of the representations learned by each method in Table A.8. Our proposed method generally maintains a higher embedding dimensionality.

Table 2: Comparison of the linear evaluation accuracy on benchmark datasets

|              | STL-10 | CIFAR-10 | CIFAR-100 | Tiny ImageNet |
|--------------|--------|----------|-----------|---------------|
| SimCLR       | 87.66  | 90.60    | 62.47     | 45.01         |
| SimSiam      | 88.91  | 90.67    | 62.49     | 50.16         |
| BYOL         | 88.05  | 85.81    | 60.69     | 52.04         |
| MOCO         | 90.36  | 87.86    | 60.36     | 40.98         |
| DirectCLR    | 86.46  | 90.08    | 60.81     | 48.01         |
| VICReg       | 89.78  | 90.55    | 62.69     | 51.01         |
| Barlow Twins | 88.36  | 89.18    | 63.47     | 49.16         |
| W-MSE        | 90.68  | 91.49    | 63.71     | 50.20         |
| FCL          | **92.52** | **91.91** | **63.76** | **55.81**  |

In addition to linear evaluation, we further use transfer learning to evaluate the performance of the trained model. We use the pre-trained model on one dataset and then evaluate the model on another dataset. The transfer learning results are shown in Table 3. The advantage of our proposed method in transfer learning is more significant when the model is pre-trained on Tiny ImageNet compared with other methods. This further validates that the pre-trained model on the Tiny ImageNet dataset by FCL can preserve more class-relevant features than other methods. The details about the transfer learning task are presented in the Appendix.

## 6 CONCLUSION

In this paper, we propose a novel contrastive learning method, FCL, to address the feature suppression effects prevalent in SSCL. Feature suppression effects, where easy-to-learn and class-irrelevant features suppress other class-relevant features, is a common problem in SSCL. Our method offers

Table 3: Comparison of the transfer learning accuracy on benchmark datasets.

| | CIFAR100 → CIFAR10 | CIFAR100 → CIFAR100 | TinyImageNet → STL-10 | TinyImageNet → CIFAR10 | TinyImageNet → CIFAR100 |
|---|---|---|---|---|---|
| SimCLR | 75.53 | 45.89 | 75.53 | 78.27 | 54.34 |
| SimSiam | 72.10 | 40.18 | 67.25 | 75.04 | 47.96 |
| BYOL | 75.09 | 41.56 | 69.43 | 79.85 | 56.54 |
| MOCO | 75.42 | 43.59 | 72.01 | 66.48 | 39.54 |
| DirectCLR | 75.10 | 42.28 | 71.38 | 76.26 | 49.89 |
| FCL | **75.64** | **46.50** | **83.32** | **81.55** | **57.98** |

a robust solution to feature suppression effects with theoretical guarantees, retaining the central sufficient dimension reduction space. We demonstrate the advantages of our method using image datasets that exhibit feature suppression effects, as well as benchmark image datasets. Compared to other self-supervised learning methods, our proposed method is more robust to various levels of feature suppression effects. When the assumption of completeness of the central dimension reduction class is not satisfied, the proposed algorithm is still unbiased but no longer exhaustive. To recover a function class that is larger than the current one, we can use the idea of the sliced average variance estimator Lee et al. (2013) when the variance of the distribution of various classes differs in self-supervised learning. Additionally, our proposed framework has broader applications in multi-modal data fusion. Instead of "CLIP"ing Radford et al. (2021) integrates two modalities like images and text using contrastive loss, the proposed framework can integrate multiple modalities using Fisher contrastive loss.

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

# Appendix for "Fisher Contrastive Learning: A Robust Solution to the Feature Suppression Effect"

The appendix shows the details of the proof, experiments including the parameter settings for generating the datasets with features suppression effects, hyper-parameters used for our proposed method and the benchmark methods, and more experiment results on datasets with feature suppression effects as well as benchmark image datasets.

## A  PROOF DETAILS

### A.1  PROOF OF LEMMA 4.2

This lemma can be easily proven by Theorem 12.2 in Li (2018).

Notice that the condition of Theorem 12.2 requires that the family of conditional probability measure

$$\{\mathbb{P}_{X|Y}(\cdot|y) : y \in \Omega_Y\} = \{\mathbb{P}_i : i = 1, \ldots, n\}$$

being dominated by a $\sigma$-finite measure, which is satisfied by our condition.

### A.2  PROOF OF THEOREM 4.6

Let $\mathcal{K}(\cdot, \cdot)$ represent the kernel function induced by the Hilbert space $\mathcal{H}_X$. Under the framework of the reproducing Kernel Hilbert space, we define the covariance operators for functions on $X$ and $Y$.

Since $Y$ is a categorical variable with $n$ classes, the domain of $Y$ can be represented by $\Omega_Y = \{1, \ldots, n\}$. Notice that the function on $Y$ can be represented by a $n$-dimensional vector, and the Hilbert space for the functions on $Y$, denoted by $\mathcal{H}_Y$, can be defined via the inner product of vectors in the Euclidean space as shown in Chapter 2.2 Gu (2013).

The variance operator for $X$ is denoted by $\Sigma_{XX}$. For a function $g : \Omega_X \to \mathbb{R}$, the operator $\Sigma_{XX}$ acts on $g$ returning a function $\Sigma_{XX}g$ having

$$\langle g, \Sigma_{XX}g \rangle_{\mathcal{H}_X} = \text{Var}[g(X)],$$

where $\langle \cdot, \cdot \rangle_{\mathcal{H}_X}$ represent the inner product in the Hilbert space $\mathcal{H}_X$.

The variance operator for $Y$ is denoted by $\Sigma_{YY}$. The operator $\Sigma_{YY}$ acts on $h$ returning a function $\Sigma_{YY}h$ having

$$\langle h, \Sigma_{YY}h \rangle_{\mathcal{H}_Y} = \text{Var}[h(Y)],$$

The covariance operator for $X$ and $Y$, $\Sigma_{XY}$ can acts on $h$ returning a function on $X$, $\Sigma_{XY}h$ having

$$\langle g, \Sigma_{XY}h \rangle_{\mathcal{H}_X} = \text{Cov}[g(X), h(Y)],$$

The covariance operator for $X$ and $Y$, $\Sigma_{YX}$ can acts on $g$ returning a function on $Y$, $\Sigma_{YX}g$ having

$$\langle \Sigma_{YX}g, h \rangle_{\mathcal{H}_Y} = \text{Cov}[g(X), h(Y)].$$

The assumptions for deriving this theorem include

**Assumption A.1.** $\text{E}[\mathcal{K}(X, X)] < \infty$

**Assumption A.2.** *The kernel of the operator* $\Sigma_{XX}$, $ker(\Sigma_{XX}) = \{0\}$, *i.e., if* $\text{Var}[f(X)] = 0$ *for* $f \in \mathcal{H}_X$, $f \equiv 0$.

**Assumption A.3.** $\text{ran}(\Sigma_{XY}) \subseteq \text{ran}(\Sigma_{XX})$, $\text{ran}(\Sigma_{YX}) \subseteq \text{ran}(\Sigma_{YY})$, *where* $\text{ran}(\cdot)$ *represent the range of the operator.*

**Assumption A.4.** *The operators* $\Sigma_{YY}^{-1}\Sigma_{YX}$ *and* $\Sigma_{XX}^{-1}\Sigma_{XY}$ *are compact.*

**Assumption A.5.** $\mathfrak{S}_{Y|X}$ *is complete.*

The result of the theorem can be derived as follows.

Based on the above assumptions and the result of Theorem 13.2 in Li (2018), we have

$$\overline{\text{ran}}(\Sigma_{XX}^{-1}\Sigma_{XY}) = \mathfrak{S}_{Y|X}.$$

From this, we have

$$\overline{\mathrm{ran}}\left(\Sigma_{XX}^{-1}\Sigma_{XY}\Sigma_{YY}^{-1}\Sigma_{YX}\Sigma_{XX}^{-1}\right) \subset \mathfrak{S}_{Y|X}$$

This suggests that we use $\overline{\mathrm{ran}}\left(\Sigma_{XX}^{-1}\Sigma_{XY}\Sigma_{YY}^{-1}\Sigma_{YX}\Sigma_{XX}^{-1}\right)$ to estimate the central $\sigma$-field $\mathfrak{S}_{Y|X}$.

This space can be recovered by sequentially solving the following problem:

$$\begin{aligned}
\underset{g_k}{\mathrm{maximize}} \quad & \left\langle g_k, \Sigma_{XY}\Sigma_{YY}^{-1}\Sigma_{XY}g_k\right\rangle_{\mathcal{H}_X} \\
\mathrm{subject\ to} \quad & \left\langle g_k, \Sigma_{XX}g_k\right\rangle_{\mathcal{H}_X} = 1, g_k \perp \mathcal{S}_{k-1}.
\end{aligned} \tag{14}$$

where $\mathcal{S}_k = \mathrm{span}\left(g_1,\ldots,g_{k-1}\right)$ and $g_1,\ldots,g_{k-1}$ are the solutions to this constrained maximization problem in the previous steps.

Since $\left\langle g, \Sigma_{XY}\Sigma_{YY}^{-1}\Sigma_{XY}g\right\rangle_{\mathcal{H}_X} = \mathrm{Var}[\mathrm{E}(g(X))|Y]$ and $\left\langle g, \Sigma_{XX}g\right\rangle_{\mathcal{H}_X} = \mathrm{Var}[g(X)]$.

Therefore, Eq.(14) is equivalent to finding a function $f : \Omega_X \to \mathbb{R}^d$, such that

$$\begin{aligned}
\underset{f}{\mathrm{maximize}} \quad & \mathrm{tr}\{\mathrm{Var}[\mathrm{E}(f(X))|Y]\} \\
\mathrm{subject\ to} \quad & \mathrm{Var}[f(X)] = \mathbf{I}_d.
\end{aligned} \tag{15}$$

With further generalization, if $\mathrm{Var}[f(X)] \neq \mathbf{I}_d$ and positive definite, solving Eq.(15) is equivalent to the following maximization problem.

$$\begin{aligned}
\underset{f}{\mathrm{maximize}} \quad & \mathrm{tr}\{\mathrm{Var}[\mathrm{E}(\mathrm{Var}[f(X)]^{-1/2}\cdot f(X))|Y]\} \\
& = \mathrm{tr}\{\mathrm{Var}[f(X)]^{-1}\cdot\mathrm{Var}[\mathrm{E}(f(X))|Y]\}.
\end{aligned} \tag{16}$$

Therefore, the unbiasedness of the solution $f_{sol} = (g_{1,sol},\ldots,g_{d,sol})$ of Eq.(16) is proven,i.e.,

Suppose there is a function $g \in \mathfrak{S}_{Y|X}$ and $g \notin \overline{\mathrm{span}}\{g_{1,sol},\ldots,g_{d,sol}\}$. Since $g_{1,sol},\ldots,g_{d,sol}$ are linear independent, we have

$$\dim(\mathfrak{S}_{Y|X}) \geq \dim(\overline{\mathrm{span}}\{g, g_{1,sol},\ldots,g_{d,sol}\}) = d+1 > \dim(\overline{\mathrm{span}}\{g_{1,sol},\ldots,g_{d,sol}\}) = d,$$

which abuse the conclusion about $\overline{\mathrm{span}}\{g_{1,sol},\ldots,g_{d,sol}\} \subset \mathfrak{S}_{Y|X}$.

Hence, we have $\mathfrak{S}_{Y|X} \subset \overline{\mathrm{span}}\{g_{1,sol},\ldots,g_{d,sol}\}$. Therefore, the exhaustiveness of the solution $f_{sol} = (g_{1,sol},\ldots,g_{d,sol})$ is proven.

# B    IMPLEMENTATION DETAILS

## B.1    MODIFIED OPTIMIZATION TARGET

In our proposed method, we target to maximize the total discrimination power $\Delta = \frac{1}{n-1}\sum_{i=1}^{n-1}\hat{v}_i$ of the proposed method. However, directly maximizing the objective could lead to trivial solutions, e.g., maximizing only the largest eigenvalue to produce the largest discriminative power. For contrastive objective, this means that it maximizes the distance of classes that are already separated at the expense of non-separated classes with less discrimination power Dorfer et al. (2015). To tackle the problem, we can modify the loss function, to maximize the smallest eigenvalues which are smaller than some threshold. The threshold is set as $\min\{\hat{v}_1,\ldots,\hat{v}_{n-1}\} + m$, where $m$ is the margin for the smallest eigenvalues to be maximized.

$$\max\sum_{i\in\Theta}\hat{v}_i \text{ with } \{\Theta\} = \{\hat{v}_j \mid \hat{v}_j < \min\{\hat{v}_1,\ldots,\hat{v}_{n-1}\} + m\}. \tag{17}$$

The intuition of the optimization goal is to increase the discrimination power as much as possible.

## C EXPERIMENTAL DETAILS

We report more specific details of the experiments, including the description of the datasets, how we split the datasets for training, data augmentation techniques we have used, hyperparameters for each method, and how we choose the hyperparameters. In addition, we report more detailed experiment results. All experiments have been conducted on a machine equipped with NVIDIA Tesla V100 GPUs (each with 32GB of memory) and a 40-core CPU (3.00 GHz).

### C.1 DESCRIPTION OF DATASETS

We first give more details about the generation of the datasets.

#### C.1.1 RANDOM BITS

The generation of the Digits on Flower dataset with strong feature suppression effect is adapted from Chen et al. (2021). In this dataset, we concatenate a real image with an image of a random integer along the channel dimension. The random integer is sampled from the range $[1, \log_2(l)]$, where $l$ is a controllable parameter. This integer is replicated across all pixel locations and expressed using $l$ binary channels. To be noted, unlike RGB channels, these additional channels of random bits are not altered by augmentation, ensuring they remain identical in both augmented views of the same image. The added channels of random bits are easy-to-learn features that suppress the other features. We vary the number of random bits, $\{0, 2, 4, 6, 8, 10\}$, in the dataset to control the amount of mutual information between two augmented views. Also, we know that the mutual information between two views given this construction is at least $\log_2(l)$. The dataset is divided into 80% training samples and 20% test samples.

#### C.1.2 DIGITS ON FLOWER DATASET

The generation of the Digits on Flower dataset with strong feature suppression effects is adapted from the method described in Chen et al. (2021). In this process, for each flower image, we map a randomly sampled MNIST digit image onto the flower image. To explore the impact of feature suppression, we vary the number of digit copies mapped onto the flower images with the following configurations: $\{0, 1, 4, 9, 16\}$ digits. The flower dataset consists of 3670 color images, each with a size of 224x224 pixels. The MNIST digit images used for mapping are resized to 72x72 pixels. For images overlaid with a single digit, the digit is placed at the center of the image. When four digits are overlaid, their positions are at $[0.3, 0.7]$ of the image dimensions. For nine digits, their positions are at $[0.25, 0.5, 0.75]$, and for sixteen digits, they are positioned at $[0.2, 0.4, 0.6, 0.8]$. The dataset is divided into 80% training samples and 20% test samples.

By varying the number and positions of the overlaid digits, we can systematically study the effect of feature suppression in the dataset. This setup allows us to control the amount of information from the digits and observe how it impacts the classification performance on the flower dataset.

The MNIST dataset LeCun (1998) is a large database of handwritten digits commonly used for training various image processing systems. It contains a total of 70,000 grayscale images, each of size 28x28 pixels. The dataset is divided into 60,000 training samples and 10,000 testing samples. Each image is labeled with one of 10 classes, corresponding to the digits 0 through 9, with approximately 7,000 samples per class. The simplicity and cleanliness of the dataset make it a standard benchmark for evaluating algorithms in self-supervised learning.

#### C.1.3 BENCHMARK DATASETS

We use four benchmark real-world datasets in SSCL to evaluate the performance of our method.

*STL-10*: The STL-10 dataset Coates et al. (2011) is designed for developing unsupervised feature learning, deep learning, and self-taught learning algorithms. It contains color images of size 96x96 pixels. The dataset includes 10 classes with 500 labeled training examples and 800 labeled testing examples per class, totaling 5,000 labeled training samples and 8,000 testing samples. Additionally, there are 100,000 unlabeled images for unsupervised learning tasks. The classes represent common objects such as airplanes, birds, and cars, making the dataset suitable for evaluating complex feature

learning algorithms. Notice that in the pretraining stage, we only use the training examples without labels.

*CIFAR-10*: The CIFAR-10 dataset Krizhevsky et al. (2009) is a widely used dataset for object recognition tasks. It consists of 60,000 color images of size 32x32 pixels, with 6,000 images per class distributed evenly across 10 classes. The dataset is split into 50,000 training samples and 10,000 testing samples. Each class represents a common object such as airplanes, cars, and birds. The dataset's diversity and balanced class distribution make it an excellent benchmark for testing self-supervised learning methods and other machine learning algorithms.

*CIFAR-100*: The CIFAR-100 dataset Krizhevsky et al. (2009) is similar to CIFAR-10 but with a greater number of classes and finer granularity. It contains 60,000 color images of size 32x32 pixels, divided into 100 classes, each with 600 images. The dataset is split into 50,000 training samples and 10,000 testing samples. Each class represents a specific object, such as a type of flower or insect, providing a more challenging task due to the increased number of classes and the fine-grained nature of the categories. This makes CIFAR-100 a comprehensive benchmark for evaluating the performance of self-supervised learning algorithms.

*TinyImageNet*: The TinyImageNet dataset is a subset of ImageNet. It contains 100,000 images of 200 classes downsized to 64×64 colored images. The dataset is split into 80,000 training samples and 20,000 testing samples. Compared with CIFAR-10, CIFAR-100 and STL-10 it has more images and more classes, which makes it a more challenging task.

## C.2 EXPERIMENTAL SETTINGS

### C.2.1 DATA AUGMENTATION

For Random Bits and Digits on Flower datasets, we only use random crop with resize Chen et al. (2020a) as the data augmentation method. For the benchmark datasets, including MNIST, STL-10, CIFAR-10, and CIFAR-100, we employ a broader range of data augmentations. These include random crop with resize, random flip, color distortion, and Gaussian blur. All processes and parameter settings are consistent with those outlined in Chen et al. (2020a) to ensure a fair comparison and reproducibility of results.

### C.2.2 MODEL SETTINGS

**Random Bits and Digits on Flower Datasets** For Random Bits and Digit on Flower datasets, the encoder includes three two-dimensional convolutional layers, followed by a Flatten layer and a Dense layer. Batch normalization and ReLU activation functions are applied throughout the encoder. The projection network consists of two Dense layers, with a final output dimension of 128. We use a dense layer for classification. Throughout the pre-training, we utilize the Adam optimizer with a polynomial decay learning rate schedule. The initial learning rate is set to 0.001, with an end learning rate of 0. The decay steps are set to 5000 for the Random Bits dataset and 800 for the Digits on Flower dataset. The number of epochs is 10 for the Random Bits dataset and 35 for the Digits on Flower dataset. The batch size is 128.

**Benchmark Datasets** For benchmark datasets, we follow the settings in SimCLR to design the architecture. The backbone network is ResNet-50 for TinyImageNet and ResNet-18 for other datasets (CIFAR10, CIFAR100 and STL10) He et al. (2016) and the projector is a two-layer MLP after ResNet's global average pooling layer ($pool_5$). Both the input and output dimensions of each layer in the projector are set to 128, with each layer followed by a ReLU activation function. During unsupervised pre-training, we use a base learning rate of 0.03 with a cosine decay schedule for 600 epochs. The weight decay is set to 0.0005, and momentum is set to 0.9. We also incorporate a warm-up phase for the first 10 epochs. The batch size is 128, and we utilize the SGD optimizer for training.

After pre-training the network, we freeze its parameters and train a supervised linear classifier in linear evaluation and transfer learning. The features used for training the classifier are extracted from ResNet's global average pooling layer ($pool_5$). For training the linear classifier, we use a base learning rate of 30 with a cosine decay schedule over 30 epochs. We also employs the SGD optimizer with a momentum of 0.9 and a batch size of 256.

### C.3 EXPERIMENTAL RESULTS

#### C.3.1 RANDOM BITS

The table A.1 presents the mean and standard deviation (in parentheses) of the accuracy (10 replications) for three different self-supervised learning methods: SimCLR, SimSiam, and our proposed FCL. The results are shown for a dataset with varying numbers of random bits introduced into the input data. Our method is the best across all the numbers of random bits. As the number of random bits increases, the performance of all three methods decreases, but FCL consistently outperforms SimCLR and SimSiam across all levels of randomness. Even with 10 random bits, FCL maintains a mean accuracy of 0.5374, while SimCLR and SimSiam both drop below 0.11.

The superior performance of FCL, especially in the presence of increasing feature suppression effects, indicates that the proposed method is more robust and can learn better representations even when feature suppression effects exist.

The left panel of Figure A.2 shows the rank patterns of the two methods. Notice that the rank for the SimCLR method varies significantly with different numbers of random bits, while the rank for our FCL method remains stable regardless of the number of random bits. Specifically, with 0, 2, 4, and 6 random bits, our rank is 127; it slightly decreases to 126 for 8 bits and to 123 for 10 bits. This steady performance by FCL highlights its reliability and efficiency.

#### C.3.2 DIGITS ON FLOWER DATASET

The table A.5 presents the mean and standard deviation (in parentheses) of the accuracy (10 replications) for three different self-supervised learning methods: SimCLR, SimSiam, and FCL (the proposed method) performed on the Digits on Flower Dataset, where varying numbers of digits are added to the flower images. When there are no digits added (0 digits), FCL achieves the highest mean accuracy of 0.5071, outperforming both SimCLR (0.4687) and SimSiam (0.3935). This suggests that FCL is better able to learn meaningful representations from the original flower images compared to the other two methods. As the number of digits added to the images increases, the performance of all three methods generally decreases, but FCL consistently outperforms SimCLR and SimSiam across all levels of digit addition. Even with 16 digits added, FCL maintains a mean accuracy of 0.4533, while SimSiam and SimCLR's accuracy drops to 0.3246 and 0.4144 respectively. The superior performance of FCL, even in the presence of increasing interference from the added digits, indicates that the proposed method is more robust and can learn better representations of the original flower images, despite the presence of easy-to-learn features (the digits). More salient maps for visualization are shown in Figure A.1

Additionally, Figure A.2 right side displays the rank of the embedding space for each method. This further confirms that our proposed method is more robust to strong feature suppression effects. In the Digits on Flower dataset, when the number of digits is set as zero, the rank of learning embedding for SimCLR and SimSim are 84 and 103 respectively, while FCL achieves a significantly higher rank of 110. As the number of digits increases, all of the methods experience a slight decrease in performance, but FCL maintains a more stable performance with rank ranging from 106 to 110, compared to SimCLR's range of 82 to 84 and SimSaim's range of 82 to 103. This consistent outperformance by FCL highlights its robustness and effectiveness.

Table A.1: Mean and Standard Deviation of Accuracy for Random Bits Dataset

| Number of Random Bits | SimCLR | SimSiam | FCL |
|:---:|:---:|:---:|:---:|
| 0 | 0.9445 (0.0019) | 0.8749 (0.0124) | **0.9790** (0.0031) |
| 2 | 0.9100 (0.0036) | 0.1368 (0.0205) | **0.9405** (0.0037) |
| 4 | 0.8130 (0.0087) | 0.1088 (0.0036) | **0.8977** (0.0073) |
| 6 | 0.3179 (0.0234) | 0.1038 (0.0036) | **0.7943** (0.0010) |
| 8 | 0.1038 (0.0153) | 0.1035 (0.0024) | **0.6314** (0.0119) |
| 10 | 0.1027 (0.0034) | 0.1017 (0.0039) | **0.5374** (0.0010) |

Table A.2: Comparison of Rank for Randbit Dataset

| Number of Random Bits | SimCLR | SimSiam | FCL |
|---|---|---|---|
| 0 | 117 | 54 | **127** |
| 2 | 120 | 5 | **127** |
| 4 | 123 | 8 | **127** |
| 6 | 124 | 11 | **127** |
| 8 | 122 | 13 | **126** |
| 10 | 120 | 16 | **123** |

Table A.3: Sensitivity Analysis of Randbits Dataset. The table shows the difference of embeddings when the various levels of noise are added to the random bits channels. Different columns correspond to various levels of Gaussian random noise with variances 0.1, 0.2, 0.3, 0.4, 0.5.

| Approach | 0.1 | 0.2 | 0.3 | 0.4 | 0.5 |
|---|---|---|---|---|---|
| SimCLR | 0 | 1.15 | 4.13 | 6.70 | 8.34 |
| SimSiam | 0 | 0.32 | 2.32 | 5.09 | 7.62 |
| FCL | 0 | 0.20 | 0.75 | 1.31 | 1.72 |

Table A.4: Sensitivity Analysis of Randbits Dataset. The table shows the difference of embeddings when the various levels of noise are added to the RGB channels. Different columns correspond to various levels of Gaussian random noise with variances of 0.1, 0.2, 0.3, 0.4, 0.5.

| Approach | 0.1 | 0.2 | 0.3 | 0.4 | 0.5 |
|---|---|---|---|---|---|
| SimCLR | 2.73 | 2.82 | 5.04 | 7.55 | 9.03 |
| SimSiam | 0.22 | 0.28 | 0.45 | 0.64 | 0.77 |
| FCL | 2.08 | 2.24 | 3.45 | 5.10 | 6.36 |

Figure A.1: Illustrations of the salient map of each method. The first column is the four-digits-on-flower image. From the second to fourth columns, salient maps of the images for each method, FCL, SimCLR, and SimSiam.

### C.3.3 BENCHMARK DATASETS

We follow the settings in C.2 for benchmark datasets. Given the pre-trained network, a supervised linear classifier is trained after ResNet-18's global average pooling layer. We report additional accuracy on MNIST LeCun (1998), STL-10 Coates et al. (2011), CIFAR-10 and CIFAR-100 Krizhevsky et al. (2009) after pre-training 200 and 400 epochs in Table A.7.

Table A.5: Mean and Standard Deviation of Accuracy for Digits on Flower Dataset

| Number of Digits | SimCLR | SimSiam | FCL |
|---|---|---|---|
| 0 | 0.4687 (0.0404) | 0.3935 (0.0402) | **0.5071** (0.0333) |
| 1 | 0.4670 (0.0439) | 0.3920 (0.0415) | **0.5015** (0.0397) |
| 4 | 0.4538 (0.0258) | 0.3669 (0.0414) | **0.4918** (0.0274) |
| 9 | 0.4388 (0.0409) | 0.3588 (0.0356) | **0.4909** (0.0283) |
| 16 | 0.4144 (0.0216) | 0.3246 (0.0214) | **0.4533** (0.0354) |

Table A.6: Comparison of Rank for Digits on Flower Dataset

| Number of Digits | SimCLR | SimSiam | FCL |
|---|---|---|---|
| 0 | 84 | 103 | **110** |
| 1 | 83 | 102 | **108** |
| 4 | 83 | 90 | **107** |
| 9 | 82 | 88 | **106** |
| 16 | 82 | 82 | **106** |

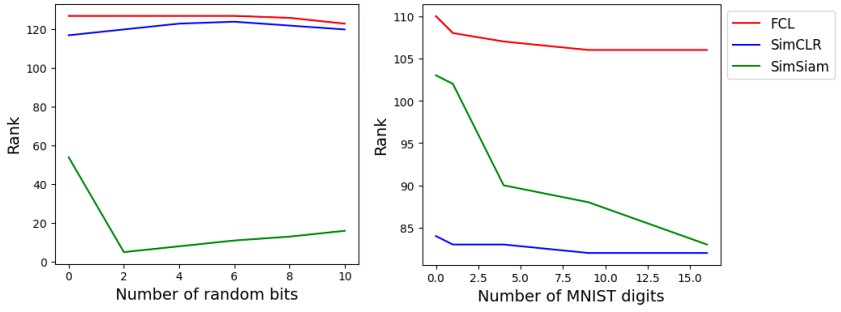

Figure A.2: Rank of learning embedding space for Randbit and Flower dataset. Left side: Randbit dataset, with number of random bits set as 0, 2, 4, 6, 8 and 10. Right side: Flower dataset, with number of digits set as 0, 1, 4, 9 and 16. Red lines: FCL method; Blue lines: SimCLR method. Green lines: SimSiam method.

Table A.7: Comparison of the linear evaluation accuracy on benchmark datasets

| Datasets | STL-10 | | CIFAR-10 | | CIFAR-100 | | Tiny-ImageNet | |
|---|---|---|---|---|---|---|---|---|
| Epochs | 200 | 400 | 200 | 400 | 200 | 400 | 200 | 400 |
| SimCLR | 66.81 | 73.09 | 83.79 | 87.78 | 54.52 | 61.09 | 43.30 | 46.46 |
| SimSiam | 60.43 | 68.04 | 86.58 | **88.97** | **59.76** | 60.10 | 43.96 | 48.01 |
| BYOL | 64.94 | 70.35 | 80.10 | 83.40 | 56.01 | 60.02 | 37.15 | 48.05 |
| MOCO | 66.46 | 72.88 | 80.96 | 82.52 | 54.27 | 58.13 | 23.08 | 33.54 |
| DirectCLR | 67.26 | 72.68 | 86.53 | 89.21 | 55.34 | 59.32 | 43.52 | 47.59 |
| FCL | **72.30** | **76.40** | **86.78** | 88.67 | 57.96 | **61.13** | **47.04** | **53.04** |

Table A.8: Embedding dimensionality, the highest is marked as **bold** and the second highest rank is underlined.

| | Dimension | SimCLR | SimSiam | BYOL | MOCO | DirectCLR | FCL |
|---|---|---|---|---|---|---|---|
| STL10 | 512 | 150 | 205 | 190 | **358** | 232 | 278 |
| CIFAR10 | 512 | 381 | 279 | 401 | 407 | 229 | **463** |
| CIFAR100 | 512 | 343 | 278 | 401 | 414 | 253 | **430** |
| TinyImageNet | 2048 | 561 | 1095 | 901 | 493 | **1571** | 1196 |

