# OpenReview forum: "Fisher Contrastive Learning: A Robust Solution to the Feature Suppression Effect"
_ICLR.cc/2025/Conference — ICLR 2025 Conference Withdrawn Submission_

### Official Review · Reviewer_t4gW · 2024-10-19

**Soundness:** 3
**Presentation:** 2
**Contribution:** 2
**Rating:** 3
**Confidence:** 4

**Summary:**

This paper aims to tackle the issue of feature suppression in self-supervised contrastive learning. The proposed method, Fisher Contrastive Learning (FCL), employs Fisher Discriminant Analysis to project data into a central sufficient dimension reduction function class for eliminating class-irrelevant features. Experimental results demonstrate that FCL outperforms SimCLR and SimSiam in mitigating the effects of feature suppression.

**Strengths:**

1. The writing and motivation are clear and easy to follow.
2. The proposed FCL sounds reasonable to solve feature suppression by eliminating class-irrelevant features.

**Weaknesses:**

1. While the proposed FCL appears conceptually sound, the experimental results do not provide sufficient evidence of its effectiveness.

- Firstly, the authors state that "using our proposed method, class-relevant features are not suppressed by strong or easy-to-learn features on datasets known for strong feature suppression effects" (L23-24). However, as shown in Figure 3, the performance of FCL degrades as the number of random bits increases. In fact, FCL deteriorates more than SimCLR when the random bits increase from 9 to 16. These results suggest that FCL still suffers from feature suppression, and its main advantage over SimCLR lies in its higher initial accuracy rather than sustained robustness.
- Secondly, a key manifestation of feature suppression is low embedding dimensionality, often referred to as dimensional or informational collapse. Methods like SimCLR and SimSiam are known to suffer from informational collapse, but there are more recent self-supervised learning approaches, such as decorrelation methods (e.g., W-MSE[1], VICReg[2], Barlow Twins[3]), which effectively mitigate informational collapse and achieve better performance in downstream tasks. I recommend that the authors compare FCL with more state-of-the-art (SOTA) methods to provide a stronger baseline.

- Thirdly, some of the experimental results appear unreliable. For instance, common contrastive self-supervised learning methods typically achieve over 90% top-1 accuracy with a linear probe on the STL-10 dataset[1,4], yet the reported accuracies in Table 2 are below 80%. Furthermore, the paper only includes experiments on small datasets, which do not sufficiently demonstrate the practical effectiveness of FCL. I suggest that the authors evaluate their method on larger datasets, such as ImageNet, to provide a more comprehensive assessment.

2. Given that self-supervised contrastive learning generally requires a large number of epochs to converge, computational efficiency is critical for practical applications. However, the paper does not provide a comparison of the computational time or resource consumption between FCL and other methods. Additionally, I noticed that FCL involves eigenvalue decomposition, which can be computationally expensive. I recommend that the authors include a detailed analysis of the method's computational cost.

3. The reproducibility of the paper is questionable. After reviewing the code provided in the supplementary materials, I found that the structure is disorganized, with inefficient practices such as using loops to traverse batches when calculating variances $S_W$ and $S_B$. I suggest that the authors revise the code structure to enhance clarity and efficiency, making it more accessible and useful for the broader research community.


[1] Aleksandr Ermolov, Aliaksandr Siarohin, Enver Sangineto, and Nicu Sebe. Whitening for selfsupervised representation learning. In ICML, 2021.

[2] Adrien Bardes, Jean Ponce, and Yann LeCun. Vicreg: Variance-invariance-covariance regularization for self-supervised learning. In ICLR, 2022.

[3] Jure Zbontar, Li Jing, Ishan Misra, Yann Lecun, and Stephane Deny. Barlow twins: Self-supervised learning via redundancy reduction. In ICML, 2021.

[4] Xi Weng, Lei Huang, Lei Zhao, Rao Muhammad Anwer, Salman Khan, and Fahad Khan. An investigation into whitening loss for self-supervised learning. In NeurIPS, 2022.

**Questions:**

- Some notations are confusing. What does $k$ represent in formula 3 ?

- What is the value of the penalty hyperparameter $\lambda$?

- As is well known, BYOL has stable performance and outperforms SimSiam and SimCLR. Why doesn't the author analyze its effectiveness in feature suppression？

**Details Of Ethics Concerns:**

N.A.

---

> ### Comment · Reviewer_t4gW · 2024-11-26
>
> Although the authors did not address or resolve my questions and concerns, the PCs have requested a discussion. I believe that this paper lacks sufficient experiments and only compares against weaker methods. It does not meet the standards required for publication at ICLR. Therefore, I will maintain my score.

---

### Official Review · Reviewer_qE5n · 2024-11-03

**Soundness:** 3
**Presentation:** 3
**Contribution:** 3
**Rating:** 6
**Confidence:** 2

**Summary:**

This paper presents a novel approach to self-supervised learning that addresses the feature suppression effect through Fisher linear discriminant analysis. The authors propose a nonlinear sufficient dimension reduction method for SSCL, which offers a robust solution to the feature suppression effects by learning an exhaustive and unbiased function class. The paper is well-written and provides a clear explanation of the proposed method, its theoretical foundation, and experimental results.

**Strengths:**

1. Novelty: The proposed FCL contributes to the field of self-supervised contrastive learning. It addresses a critical problem, the feature suppression effect, with a theoretically sound approach.

2. Theoretical Guarantee: The paper provides a solid theoretical foundation for the proposed FCL, demonstrating that it can learn the central sufficient dimension reduction subspace for group separation.

3. Empirical Evaluation: The experimental results on datasets with feature suppression effects and benchmark image datasets demonstrate the effectiveness of FCL compared to existing methods.

4. Ease of Execution: The code is easy to execute and requires minimal dependencies.

**Weaknesses:**

Computational Complexity: The paper does not discuss the computational complexity of FCL. It would be beneficial to analyze and compare the computational efficiency of FCL with other SSCL methods.

I reviewed the code and noted that the execution process is not complex. However, in my opinion, an analysis of the computational complexity of the proposed FCL is necessary to demonstrate the superiority of the method.

**Questions:**

See the Weaknesses, namely the computational complexity of FCL.

---

### Official Review · Reviewer_nf8D · 2024-11-03

**Soundness:** 3
**Presentation:** 2
**Contribution:** 3
**Rating:** 3
**Confidence:** 2

**Summary:**

This paper propose a novel contrastive learning method, Fisher Contrastive Learning, to address feature suppression effects in SSCL. The paper provides mathematical theories and proofs of FCL as theoretical basis for the method. Under a certain condition, FCL can perform well in theory. In practical scenarios, the authors conduct related experiments to identify their method, which achieves better results than some of other SSCL methods, in some datasets.

**Strengths:**

- The motivation is clear, and the writing is easy to follow.
- FCL arises from FDA, it is insightful to apply the classic idea to recent research.
- The mathematical descriptions and theorems seem to provide strong support for this paper.

**Weaknesses:**

My mild concern is about the descriptions of mathematics.

From my personal perspective, although I am also a theory researcher in deep learning and have completed lots of mathematical courses. I am still not familiar with some of the mathematical signs, especially in **Section 4**. I think most readers in artificial intelligence rather in mathematics may have the same confusions when reading this paper.

For example, when I read **Definition 4.1** and **Lemma 4.2**, I feel unfamiliar with these signs. I thus decide to refer to the appendix for the details to know more about them, but I find that I the proof is omitted---I need to refer to other papers or books to understand the concept in this paper. I think this is not convenient for most readers to refer much more other papers when reading this paper, unless this paper surely requires high-level mathematics.

**Questions:**

For **section 4**, could the authors please explain why these theorems are important in FCL? What is the advantage of FCL than other methods from these theorems?

---

> ### Comment · Reviewer_nf8D · 2024-11-26
>
> Thanks for adding the necessary mathematical descriptions for better understanding.
>
> In the reply to Reviewer qE5n, I noticed that "The computation cost of eigenvalue decomposition and cosine similarity are on the same scale when $d=n$". Could the authors please give the further demonstrations that why $O(kd^2)$ is equal to $O(dn^2)$, or my understanding is not right?
>
> On the other hand, I noticed that the authors did not address Reviewer t4gW's concerns, who seems to have a deeper understanding of the field. Additionally, I observed that the authors added a "Comparison of the computational time" section at the end of the paper. However, I find it hard to believe that such a complex method could achieve a training speed three times faster than SimSiam and SimCLR. Therefore, I have decided to lower my score to 3 temporarily.

---

### Note · Authors · 2025-01-01

I have read and agree with the venue's withdrawal policy on behalf of myself and my co-authors.